# Bootstrapping Small & High Performance Language Models with Unmasking-Removal Training Policy

**Yahan Yang[1], Elior Sulem[2], Insup Lee[1], Dan Roth[1]**
[1] Department of Computer and Information Science, University of Pennsylvania
[2] Department of Software and Information Systems Engineering,
Ben-Gurion University of the Negev
{yangy96, lee, danroth}@seas.upenn.edu
eliorsu@bgu.ac.il

## Abstract

BabyBERTa, a language model trained on small-scale child-directed speech while none of the words are unmasked during training, has been shown to achieve a level of grammaticality comparable to that of RoBERTa-base, which is trained on 6,000 times more words and 15 times more parameters (Huebner et al., 2021). Relying on this promising result, we explore in this paper the performance of BabyBERTa-based models in downstream tasks, focusing on Semantic Role Labeling (SRL) and two Extractive Question Answering tasks, with the aim of building more efficient systems that rely on less data and smaller models. We investigate the influence of these models both alone and as a starting point to larger pre-trained models, separately examining the contribution of the pre-training data, the vocabulary, and the masking policy on the downstream task performance. Our results show that BabyBERTa trained with unmasking-removal policy is a much stronger starting point for downstream tasks compared to the use of RoBERTa masking policy when 10M words are used for training and that this tendency persists, although to a lesser extent, when adding more training data. [1]

## 1 Introduction

Large-scale pre-trained language models (LMs) (Devlin et al., 2019; Liu et al., 2019; Yang et al., 2019) have shown promising ability on handling various downstream tasks including textual classification (Wang et al., 2018) and question answering (QA, Rajpurkar et al., 2016). Previous research (Zhang et al., 2021; Warstadt et al., 2020) showed that the size of architecture and amount of pre-training data actually affect the linguistics features learned by state-of-the-art (SOTA) pre-trained LMs like RoBERTa (Liu et al., 2019). It also showed LMs require much more data to understand com-

monsense knowledge in order to achieve high performance on natural language understanding tasks, compared to grammatical ability. On the other hand, LMs like RoBERTa are costly to train in terms of GPU computation power and time at both pre-training and fine-tuning stages. Huebner et al. (2021) proposed BabyBERTa, a smaller RoBERTa architecture that is trained on a 5M child-directed data corpora without using unmasked tokens during the masked language modeling training. BabyBERTa reaches the same level of grammaticality as RoBERTa but considerably saves training expenses. However, no further evaluation on tasks other than the grammaticality tests were performed for the model. Deshpande et al. (2023) also focus on the performance of smaller language models but emphasize the relationship between the architecture size and downstream task performance and train on larger data corpora. In this paper, we would like to answer the following questions: (1) *What is the performance for LMs based on smaller models like BabyBERTa on downstream tasks that require fine-tuning?* and (2) *What is an efficient way to improve the behavior of small pre-trained LMs on downstream tasks?*

In our work, we first evaluated both BabyBERTa and RoBERTa on three downstream tasks that target sentence structure and are closely associated with grammatical capabilities. Additionally, we propose to have various starting points by combining different ingredients in pre-training including the masking policy, the size of the vocabulary, and the type of the pre-training data corpus (child-directed language, online written language). Then, we continually pre-train BabyBERTa and its variants on more Wikipedia data to improve performance on target tasks. We observe that: (1) although BabyBERTa has a lower performance on the downstream tasks compared to RoBERTa, the use of the unmasking removal policy and of a small vocabulary is still effective after fine-tuning; (2)

---

[1] Our code can be found in https://github.com/yangy96/babyberta_continual

running thorough experiments to identify which factors are important for small language models when performing continual pre-training, we find that the influence of the unmasking removal policy persists, although to a lesser extent, when adding more training data.

## 2 Background

### 2.1 Masked language model objective (MLM)

The large transformers for language models are pre-trained on billions of tokens and show their high-capability in various downstream tasks. The success of large-scale pre-trained language models is inseparable from the masked language model objective, which is a widely-used self-supervised learning approach to construct a text representation (Devlin et al., 2019). With the MLM objective, there are $p\%$ of the tokens that are masked, and the model learns to reconstruct the masked tokens at the pre-training stage. The loss function is defined as

$$\mathcal{L} = -\sum_{i=1}^{n}\sum_{j=1}^{m_i} \log P(w_{i,j}|\tilde{x}_i) \quad (1)$$

where $w_{i,j}$ is the ground truth of the $j$th masked tokens of $i$th sequence and $\tilde{x}_i$ is the masked context, $n$ is the total number of sentences and $m_i$ is the number of masked tokens for the sentence.

#### 2.1.1 80-10-10 Masking Policy

80% of the masked tokens are replaced by the <mask> token, 10% are replaced by randomly selected tokens, and the rest are kept as the same tokens (Devlin et al., 2019). In the paper, we use the 80-10-10 and RoBERTa masking policy interchangeably.

#### 2.1.2 Unmasking Removal Policy

Different from the default masking strategy, we instead remove the prediction of unchanged / unmasked tokens. In other words, we replace corrupted tokens with <mask> 90% of the time and use random tokens 10% of the time[2]. Previous work (Huebner et al., 2021; Wettig et al., 2023) shows masking policies are important to pre-training.

### 2.2 BabyBERTa

BabyBERTa (Huebner et al., 2021), a small-scale version of RoBERTa, differs from RoBERTa in architecture, corpora, masking policy, and other pre-

training hyperparameters. The details are shown in Table 8. The default masking policy of Baby-BERTa is the unmasking removal policy, and the pre-training data corpora is AO-CHILDES (Huebner and Willits, 2021), which consists of child-directed speech. We consider four versions of BabyBERTa, each of them being trained on a different corpus (Table 1).

|  | Corpora |
| --- | --- |
| BabyBERTa-CHILDES | CHILDES (child-directed speech) |
| BabyBERTa-Wikipedia | Wikipedia (a small subset of Wikipedia dataset) |
| BabyBERTa-Curriculum | Combines CHILDES, Newsela, and Wikipedia |
| BabyBERTa-Combined | Combines two Wikipedia subsets of the same size |

Table 1: Data corpora for training BabyBERTa and its three variations, where CHILDES and Wikipedia contain the same number of sentences.

### 2.3 Fine-tune on downstream tasks

In this work, we are interested in following downstream tasks including semantic role labeling (SRL) and two extractive question-answering tasks: question-answer driven semantic role labeling (QASRL) and question-answer meaning representation (QAMR).

1) **SRL** Semantic role labeling aims to detect predicates (in most cases, verbs) in a sentence and assign its associated arguments with different semantic roles (Palmer et al., 2010; Carreras and Màrquez, 2005; He et al., 2017). In this paper, we evaluate models on CoNLL12, an SRL benchmark based on OntoNotes v5.0 dataset (Pradhan et al., 2013).

2) **QASRL** (He et al., 2015) also presents the predicate-argument structure in the sentence but in the format of question-answer pairs [3]. In this paper, we evaluate models on the QA-SRL Bank 2.1 dataset (FitzGerald et al., 2018).

3) **QAMR** (Michael et al., 2018) provides predicate-argument structure for more diverse relationships compared to those presented in QASRL and SRL (including noun relationship).

## 3 BabyBERTa on downstream tasks

In this section, we evaluate BabyBERTa models on various downstream tasks and experiment with different methods including continually pre-training. To perform question-answering tasks like QAMR and QASRL, we train two linear layers on top of

---

[2] We use 90-10 masking policy and unmasking removal policy interchangeably

[3] We here address the Extractive QA tasks (rather than the parsing tasks) related to the QAMR and QASRL formalisms.

the encoder of the language model (LM) to predict the start and end of the answer span within the context. We implement the classifier using Huggingface (Wolf et al., 2020). For fine-tuning LMs for SRL tasks, we utilize the implementation provided in (Zhang et al., 2022).

## 3.1 How does the BabyBERTa perform on downstream tasks?

We are interested in the performance of BabyBERTa and its variations on downstream tasks including SRL, QASRL, and QAMR. We report the F1 score in Table 2, and compare the performance of BabyBERTa models and RoBERTa. Our experiments show that BabyBERTa has comparable performance on QASRL with RoBERTa-10M and only 3 points lower compared to RoBERTa. For tasks like SRL and QAMR, BabyBERTa's performance is also within a slight 3-point margin in comparison to RoBERTa-10M. We also observe that the content of the pre-training dataset impacts its performance on downstream tasks. The Wikipedia dataset is closer to the target domain compared to the other two datasets, so BabyBERTa pre-trained on Wikipedia dataset achieves higher performance on QAMR, which is a more challenging task.[4]

| Pre-trained Models | SRL | QASRL | QAMR |
|---|---|---|---|
| BabyBERTa-CHILDES | 72.38 | 87.57 | 54.03 |
| BabyBERTa-Wikipedia | 75.96 | 90.09 | **77.43** |
| BabyBERTa-Curriculum | **77.89** | **90.13** | 73.88 |
| BabyBERTa-Combined | 76.17 | 89.9 | 77.05 |
| RoBERTa-10M | 79.75 | 90.44 | 80.76 |
| RoBERTa | 85.00 | 93.11 | 90.58 |

Table 2: Performance (F1-score) of BabyBERTa and its variants on three different downstream tasks. The performance of RoBERTa and RoBERTa-10M serves as a baseline to compare.

### 3.1.1 Effect of vocabulary size

The vocabulary size of RoBERTa is approximately 6x that of BabyBERTa, so it is possible that the size of the vocabulary size limits the understanding of language at the MLM training stage. In this experiment, we compare the performance with different vocabulary sets for pre-training the BabyBERTa model. Table 3 summarizes our experiments for pre-training BabyBERTa with various factors. We observe that the larger vocabulary does not give any improvement in most of the cases. We hypothesize that the training efficiency is low for Baby-

---

[4]More details about the training procedure are in Appendix.

BERTa when we have a larger vocabulary but less pre-training data.

### 3.1.2 Effect of masking policy

One observation in previous work (Huebner et al., 2021) is that, compared to BabyBERTa trained with 80-10-10 masking policy, BabyBERTa trained with unmasking-removal policy achieves higher scores on grammar tests. This leads to an interesting questions: *what is the impact of the masking policy of the starting point on downstream tasks?* Here, we apply two masking policies at the pre-training stage. The results in Table 3 show that the unmasking policy works better for models with smaller architectures like BabyBERTa on these three downstream tasks. Thus, we conclude that BabyBERTa pre-training with unmasking removal policy and smaller vocabulary set achieves the best performance across three different tasks given the results in Table 3.

## 3.2 Does continually pre-training BabyBERTa improve downstream tasks performance?

Since there is a performance gap between Baby-BERTa and RoBERTa as shown in previous experiments, we consider improving the performance by continually pre-train the BabyBERTa architecture on more data. To be specific, each time we pretrain the models on a new subset of the Wikipedia dataset contains about 100M words repeatedly. Given the results in section 3.1, we choose the starting points[5] trained with the unmasking removal policy and BabyBERTa vocabulary set.

For all continually pre-train procedures, we keep using RoBERTa masking policy. The masking ratio used in our experiments is 15% as the default. Table 4 presents the downstream performance of models trained with continual pre-training, considering various starting points.[6] For comparison, we include results from RoBERTa-100M (Zhang et al., 2021). To assess the impact of a more diverse dataset, we mix BookCorpus and Wikipedia as an additional dataset for continual pre-training (Gururangan et al., 2020) [7]. We observe that the smaller architecture, after continually training on 100M data, can achieve better and comparable performance for the QASRL and QAMR tasks respectively, compared to a RoBERTa-base pre-trained

---

[5]A starting point here is a BabyBERTa model with a specific masking policy, vocabulary set, and initial training corpus

[6]The results correspond to the mean value of three runs.

[7]Mixed with a ratio 1:3 as in (Zhang et al., 2021)

| BabyBERTa-CHILDES | | | | | BabyBERTa-Wikipedia | | | | |
|---|---|---|---|---|---|---|---|---|---|
| URP$_S$ | Vocabulary | SRL | QASRL | QAMR | URP$_S$ | Vocabulary | SRL | QASRL | QAMR |
| yes | RoBERTa | 69.47 | 87.19 | 53.72 | yes | RoBERTa | 74.41 | 89.94 | 69.61 |
| no | RoBERTa | 70.03 | 86.54 | 53.57 | no | RoBERTa | 73.53 | 89.52 | 66.26 |
| yes | BabyBERTa | 72.38 | **87.57** | **54.03** | yes | BabyBERTa | **75.96** | **90.09** | **77.43** |
| no | BabyBERTa | **72.44** | 86.72 | 53.36 | no | BabyBERTa | 75.86 | 89.13 | 68.7 |
| BabyBERTa-Combined | | | | | BabyBERTa-Curriculum | | | | |
| URP$_S$ | Vocabulary | SRL | QASRL | QAMR | URP$_S$ | Vocabulary | SRL | QASRL | QAMR |
| yes | RoBERTa | 73.59 | 89.36 | 67.61 | yes | RoBERTa | 73.38 | 89.18 | 66.61 |
| no | RoBERTa | 73.11 | 89.25 | 66.17 | no | RoBERTa | 73.10 | 88.99 | 64.12 |
| yes | BabyBERTa | **76.17** | **89.9** | **77.05** | yes | BabyBERTa | 77.89 | 90.13 | **74.96** |
| no | BabyBERTa | 76.13 | 89.84 | 69.23 | no | BabyBERTa | 75.81 | 89.61 | 66.84 |

Table 3: Performance (F1-score) of BabyBERTa and its variants on three different downstream tasks. We evaluate the impact of the vocabulary size and the masking policies. We highlight the best performance for different pre-training corpora. URP$_S$ indicates whether the Unmasking Removal Policy (URP) is used during pre-training.

on 10M data. Moreover, among the small models, the BabyBERTa-Wikipedia model trained on the mixed dataset overall demonstrates the best performance on QAMR and SRL, and achieves comparable performance on QASRL with the best-performing model.

| Model | Dataset | SRL | QASRL | QAMR |
|---|---|---|---|---|
| Wiki | wiki | 78.18 | 90.73 | 79.98 |
|  | mixed | **78.47** | 90.73 | **80.29** |
| Comb | wiki | 78.14 | 90.63 | 79.87 |
|  | mixed | **78.47** | 90.60 | 79.44 |
| Curr | wiki | 78.47 | 90.68 | 79.61 |
|  | mixed | 78.33 | **90.75** | 79.50 |
| RoBERTa-10M | mixed | 79.75 | 90.44 | 80.76 |
| RoBERTa-100M | mixed | 80.31 | 91.82 | 87.24 |
| RoBERTa | mixed | 85.00 | 93.11 | 90.58 |

Table 4: Downstream performance of continually pre-train BabyBERTa on more data. The models are trained on 100M tokens in total. (The starting points are using BabyBERTa vocabulary set and 90-10 masking policy.) We highlight the best performance across different starting points and datasets.

Additionally, we show how the masking policy of the starting points affects continually pre-training in Table 5. We report the mean value of three runs of the models in the table and use the methods recommended in (Dror et al., 2018) for F1 score evaluation: we apply bootstrap to perform the significance test of 3 runs and get the $p$-value of 0.04 when $\alpha = 0.05$ for BabyBERTa-CHILDES and 0.0 for BabyBERT-Wikipedia. It again shows that the BabyBERTa-CHILDES and BabyBERTa-Wikipedia gain from unmasking removal policy for QAMR. We summarize that for BabyBERTa-CHILDES and BabyBERT-Wikipedia, the unmasking removal policy at the starting point improves the performance on downstream tasks, and, for QAMR, even after continuing pre-training. However, the BabyBERTa-Curriculum does not show the same trend on QAMR ($p = 0.25$).

| Model | URP$_S$ | URP$_C$ | SRL | QASRL | QAMR |
|---|---|---|---|---|---|
| CHIL | no | no | 78.04 | 90.48 | 77.60 |
|  | yes | no | 78.08 | 90.43 | 77.88 |
|  | yes | yes | **78.19** | **90.56** | **78.60** |
| Wiki | no | no | 77.95 | 90.40 | 74.83 |
|  | yes | no | 78.07 | 90.78 | 79.88 |
|  | yes | yes | **78.08** | **90.93** | **80.43** |
| Curr | no | no | 77.93 | 90.64 | 79.57 |
|  | yes | no | 78.22 | 90.67 | 79.6 |
|  | yes | yes | **78.27** | **90.77** | **79.68** |

Table 5: The impact of masking policy after pre-training on more data. The models are pre-trained on 100M tokens in total. URP$_S$ indicates whether the Unmasking Removal Policy (URP) is used at the starting point and URP$_C$ denotes whether URP is used for continually pre-training. CHIL, Wiki, and Curr refer to BabyBERTa-CHILDES, Wikipedia and Curriculum respectively. We highlight the best performance for each starting points.

## 3.3 Scale to more data

After combining the optimal training policies as discussed in the previous section, we continually pre-train the smaller models on more data. The learning curve of the model is presented in Figure 1 on downstream tasks as more data become available (500M tokens). The performance continually improves as we keep pre-training the model on new data sequentially. In Table 6, we report the final performance after continually pre-training the model on 1B tokens. However, the performance is still lower than that of RoBERTa-base (Liu et al., 2019).

| Model | SRL | QASRL | QAMR |
|---|---|---|---|
| Comb | 79.40 | 91.29 | 82.37 |
| RoBERTa | 85.00 | 93.11 | 90.58 |

Table 6: Performance (F1-score) of continually pre-training BabyBERTa with 1B tokens on three different downstream tasks.

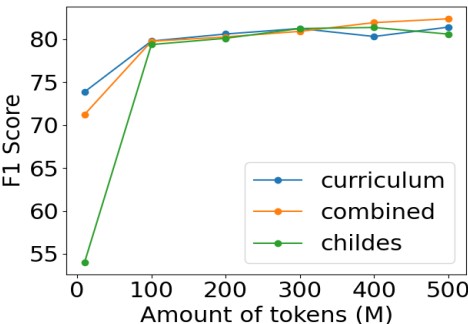

Figure 1: F1 score versus the size of tokens. Continually pre-training various variants of BabyBERTa until 500M tokens. (On QAMR task)

## 4 Conclusion

In this work, we investigate three important factors for improving smaller language models on downstream tasks: vocabulary set, masking policy, and dataset at the starting point. Our findings reveal that continuously pre-training a smaller model like BabyBERTa leads to continued improvement in downstream performance. Additionally, employing the unremoval masking policy and utilizing a smaller vocabulary prove advantageous for downstream tasks. We provide a comprehensive investigation into the relationship between pre-training procedures and downstream tasks for small models. In future research, we aim to delve deeper into the abilities acquired during the pre-training stage and their impact on downstream task performance.

## Limitations

Our study specifically concentrated on masked language model objectives and downstream tasks that are closely associated with grammaticality. However, it would be interesting to evaluate our findings on diverse downstream tasks, such as the GLUE benchmark (Wang et al., 2018). Furthermore, our investigation primarily focused on the BabyBERTa architecture configuration and small data corpus ($\leq$ 1B). It would be valuable to explore the correlation between different pre-training factors and various architecture configurations.

## Acknowledgements

We thank the members of the Cognitive Computation Group and the anonymous reviewers for their insightful suggestions. Research was sponsored by the Army Research Office and was accomplished under Grant Number W911NF-20-1-0080. It was also supported by Contracts FA8750-19-2-0201 and FA8750-19-2-1004 with the US Defense Advanced Research Projects Agency (DARPA) as well as by grants from the Israeli Ministry of Innovation, Science & Technology (#000519) and the BGU/Philadelphia Academic Bridge (The Sutnick/Zipkin Endowment Fund). Approved for Public Release, Distribution Unlimited. The views expressed are those of the authors and do not reflect the official policy or position of the Army Research Office, the Department of Defense or the U.S. Government. The U.S. Government is authorized to reproduce and distribute reprints for Government purposes notwithstanding any copyright notation herein. This research was also supported by a gift from AWS AI for research in Trustworthy AI.

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

# A  Appendix

## A.1  Dataset details

Here we provide more details about the dataset of the downstream tasks.

|       | Train  | Validation | Test  |
|-------|--------|------------|-------|
| SRL   | 75187  | 9603       | 9479  |
| QASRL | 215427 | 38487      | 45387 |
| QAMR  | 50509  | 18772      | 18596 |

Table 7: The number of samples in SRL, QASRL, and QAMR datasets.

## A.2  Architecture and datasets of BabyBERTa

Here we provide the model configurations and the size of the datasets.

|                   | RoBERTa | BabyBERTa |
|-------------------|---------|-----------|
| layers            | 12      | 8         |
| attention heads   | 12      | 8         |
| hidden size       | 768     | 256       |
| intermediate size | 3072    | 1024      |
| vocabulary size   | 50265   | 8192      |

Table 8: Architecture of BabyBERTa and RoBERTa.

|                       | Dataset Size |
|-----------------------|--------------|
| BabyBERTa-CHILDES     | 6.5M         |
| BabyBERTa-Wikipedia   | 15.91M       |
| BabyBERTa-Curriculum  | 31.81M       |
| BabyBERTa-Combined    | 31.92M       |

Table 9: Data corpora size for training BabyBERTa and its three variations.

## A.3  Compare masking policy with more pre-training data

In this section, we investigate the impact of masking policy of starting point when continually pre-training the model with more than 100M tokens. Specifically, we plot the performance on QAMR versus the number of tokens for BabyBERTa-CHILDES trained with unremoval masking policy and 80-10-10 masking policy. We observe that the performance of CHILDES trained with unremoval masking policy keeps getting better performance compared to CHILDES with 80-10-10 masking policy after continue pre-training on more and more data.

## A.4  Continually pre-train on the task-specific data

Prior work (Gururangan et al., 2020) suggests that continually pre-training on a task-specific dataset is an effective domain adaptation for downstream tasks. Following this work, we continually pre-training the model on the dataset such as QASRL,

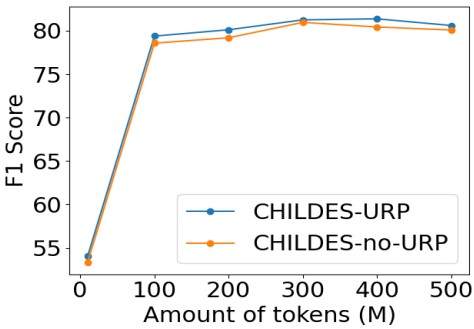

Figure 2: F1 score versus the size of tokens to compare the . Continually pre-training various variants of Baby-BERTa until 500M tokens. (On QAMR task)

QAMR, and OntoNotes. To ensure consistency, we convert the context from QASRL and QAMR into the same format as the pre-training data. The results are listed in Table 10.

| Model | Dataset | SRL | QASRL | QAMR |
|-------|---------|------|--------|-------|
| Wiki | N/A | 76.01 | **90.09** | 77.43 |
| | QAMR | 76.1 | 90.03 | 77.47 |
| | QASRL | 76.01 | 89.92 | 77.73 |
| | OntoNotes | **77.16** | 90.07 | **77.78** |
| Comb | N/A | 76.17 | 89.9 | 77.05 |
| | QAMR | 76.84 | **90.17** | 77.91 |
| | QASRL | 76.43 | 89.92 | 77.74 |
| | OntoNotes | **77.25** | 90.03 | **78.13** |
| Curr | N/A | 76.11 | 89.75 | 74.96 |
| | QAMR | **76.13** | 89.88 | 76.52 |
| | QASRL | 75.97 | 89.61 | 75 |
| | OntoNotes | 76.7 | **89.94** | **76.02** |

Table 10: Performance of continually pre-train Baby-BERTa on task-specific data.

## A.5 Continually pre-train with RoBERTa vocabulary

Here we present additional results of continually pre-train the model with RoBERTa vocabulary.

| Model | Vocab | QASRL | QAMR |
|-------|-------|--------|-------|
| Wiki | BabyBERTa | 90.77 | 80.03 |
| | RoBERTa | 90.93 | 80.38 |
| Comb | BabyBERTa | 90.78 | 79.76 |
| | RoBERTa | 90.94 | 78.38 |
| Curr | BabyBERTa | 90.72 | 79.76 |
| | RoBERTa | 91.05 | 76.91 |

Table 11: Downstream performance of continually pre-train BabyBERTa on more data for different vocabulary sets. The models are trained on 100M tokens in total. (The starting points are using 90-10 masking policy.)

## A.6 Implementation details

All of our models are trained and evaluated on two Nvidia Quadro RTX 6000. At the initial pre-training stage, the number of steps we use is 260K and the batch size is 16. The learning rate is 1e-4 and the weight decay is 0.01. At the continually pre-training stage, the number of steps we use is 300K and the batch size is 256. The learning rate is 1e-4, the warmup steps are set to be 6000 and the weight decay is 0.01.

At the fine-tuning stage for QAMR, QASRL and SRL, the model is fine-tuned on the target dataset for 10 epochs, 3 epochs, and 10 epochs respectively. The batch size is set to 16 and the learning rate is 2e-4.