# OpenReview forum: "Bootstrapping Small \& High Performance Language Models with Unmasking-Removal Training Policy"
_EMNLP/2023/Conference — EMNLP 2023 Main_

### Official Review · Reviewer_oY1d · 2023-08-04

**Soundness:** 3

**Excitement:**

3: Ambivalent: It has merits (e.g., it reports state-of-the-art results, the idea is nice), but there are key weaknesses (e.g., it describes incremental work), and it can significantly benefit from another round of revision. However, I won't object to accepting it if my co-reviewers champion it.

**Paper Topic And Main Contributions:**

This paper empirically evaluates the impact of three factors for improving small language models on downstream tasks: the pretraining data, the vocabulary, and the masking policy. Three downstream tasks are evaluated: semantic role labeling (SRL), question answer driven semantic role labeling (QASRL), and question answer meaning representation (QAMR). The evaluations reveal that unremoval masking policy can generally improve model performance, while increasing vocabulary size does not bring positive impact. Besides this, continuously pretraining the small LM with more data leads to continued improvements.

**Questions For The Authors:**

Line 236: how about the other combinations besides BabyBERTa-CHIL with and without URP on QAMR? Do we consistently see improvements with URP with statistical significance?

**Reasons To Accept:**

This paper presents valuable insights into the pretraining/finetuning of small language models. This could have a significant impact, as small language models are much easier to pretrain and finetune than larger models.

**Reasons To Reject:**

As an empirical analysis paper, the small LM’s performance in the experiments is only evaluated on two kinds of downstream tasks (SRL and two extractive question answering). It would be more convincible to conduct the evaluation on more tasks with broader coverage.

**Reproducibility:**

4: Could mostly reproduce the results, but there may be some variation because of sample variance or minor variations in their interpretation of the protocol or method.

**Reviewer Confidence:**

2: Willing to defend my evaluation, but it is fairly likely that I missed some details, didn't understand some central points, or can't be sure about the novelty of the work.

---

> ### Author Rebuttal · Authors · 2023-08-28
>
> Thank you for your valuable comments.
> 1. The reason for evaluating the performance of BabyBERTa on SRL-based downstream tasks is their focus on sentence structure analysis and their close relationship with grammaticality where BabyBERTa shows superior performance. Evaluating its performance on broader NLU tasks like the GLUE dataset goes beyond the scope of this paper. However, leveraging the strengths of smaller language models like BabyBERTa for broader tasks is an interesting future direction. We believe this paper is a first step to analyze how small language models can be used for downstream tasks.
>
> Q1. Yes, we also tried other configurations with/without URP and showed that smaller models benefit from URP for QAMR. Here, we attach the results of BabyBERTa-Wikipedia (starting point trained on Wikipedia dataset) and observe that improvements with URP is 5 points higher for this model. We will include these results in the final version of the paper.
>
>  |        | URP\_S  | URP\_C | QAMR           |
> |-----| ----| ----| ----|
> |BabyBERTa-Wiki | no     | no     | 75.61          |
>  |                             | yes    | no    | 80.09    |
>   |                            | yes    | yes    | **80.23** |

---

### Official Review · Reviewer_rmr4 · 2023-08-07

**Soundness:** 3

**Excitement:**

3: Ambivalent: It has merits (e.g., it reports state-of-the-art results, the idea is nice), but there are key weaknesses (e.g., it describes incremental work), and it can significantly benefit from another round of revision. However, I won't object to accepting it if my co-reviewers champion it.

**Paper Topic And Main Contributions:**

This work is built upon BabyBERTa (a smaller version of RoBERTa-base trained on small-scale corpus).
- While the original paper evaluates the linguistic knowledge with probing tasks based on minimal pairs, this work evaluates BabyBERTa on three grammar-related downstream tasks (SRL/CoNLL-12, QASRL, QAMR) within the pretrain-and-finetune paradigm.
- They analyze the effect of pretraining factors (corpus, vocabulary size, unmasking removal) on the downstream performance.
- They further investigate the pretrain-continue-finetune paradigm. They continually pretrain on 100M Wikipedia data and analyze the unmasking removal strategy.

**Reasons To Accept:**

- It evaluates BabyBERTa in the downstream tasks: BabyBERTa has comparable performance with RoBERTa in QASRL; there is a noticeable performance gap in SRL and QAMR. The gap can be mitigated to some extent by continual training.

- It analyzes the effect of the unmasking-removal strategy in pretraining BabyBERTa: the removal is beneficial with small-size pretrained data (5M) and the advantage diminishes when the model is continually trained with more data (100M). It may inspire future work to consider unmasking-removal in low-resource pretraining (small data, small model).


**Reasons To Reject:**

- Incremental work: The prior work BabyBERTa extensively explores the pretraining strategies and hyperparameters, and finds that unmasking-removal in pretraining is effective for syntactic probing tasks. This work validates the finding in the SRL-based downstream tasks.

- Limited applicability: while the work highlights unmasking-removal, it only experiments with very specific architecture (BabyBERTa) and with three syntactic tasks. Further, in this setup, the effectiveness holds within a narrow range (data size 5M-100M).


**Reproducibility:**

3: Could reproduce the results with some difficulty. The settings of parameters are underspecified or subjectively determined; the training/evaluation data are not widely available.

**Reviewer Confidence:**

3: Pretty sure, but there's a chance I missed something. Although I have a good feel for this area in general, I did not carefully check the paper's details, e.g., the math, experimental design, or novelty.

**Typos Grammar Style And Presentation Improvements:**

- Appendix Table 8: the header "BabyBERTa" and "RoBERTa" should be exchanged.

- line 115: hyperparamters -> hyperparameters

- I cannot understand Sec 3.1 Line 162-169. ``Our experiments show that BabyBERTa has comparable performance on QASRL and SRL compared to the other two tasks since QASRL and SRL are only points lower compared to RoBERTa’s performance''. From Table 2, BabyBERTa is indeed only 3 points lower than RoBERTa in QASRL, but there is a noticeable gap for SRL.

- Table 1: What is the difference between BabyBERTa-Wikipedia and BabyBERTa-Combined? wiki1 vs. wiki1 + wiki2? How large is each dataset?

---

> ### Author Rebuttal · Authors · 2023-08-28
>
> Thank you for your valuable comments.
> 1. Incremental work: Previous work of BabyBERTa only concentrated on the zero-shot evaluation on grammar test suites while our work focuses more on its performance on downstream tasks in the pretrain and then fine-tune paradigm. Our work shows that the small language model, trained on child acquisition language, benefits from unmasked removal policy and smaller vocabulary for downstream tasks like SRL. We also demonstrate BabyBERTa can be used as a starting point and its performance on downstream tasks can be enhanced through continuous training. We consider this work to be an important step in extending the utility of BabyBERTa and contributing to the understanding of small language models.
> 2. Limited Applicability: As we wrote in the Limitations section, we agree that more ablation study on model architecture and more tasks would be an interesting future directions to deepen the understanding of unremoval masking policy and small language models. Nonetheless, the primary focus of this paper is centered on comprehending the utilization of a small language model like BabyBERTa for structural downstream tasks and examining strategies such as unremoval masking policy to enhance its performance. The advantage of larger pretraining data corpus (over 100M) is out of the scope of this paper.
>
> Presentation improvements:
> - 1 & 2. Thank you for pointing this out. We will fix the typos in Table 8 and line 115.
> - 3. We are sorry about the confusion of the sentences line 162-169. The revised version should be  “Our experiments show that BabyBERTa has comparable performance on QASRL with RoBERTa-10M and only 3 points lower compared to RoBERTa. For tasks like SRL and QAMR, BabyBERTa's performance is also within a slight 3-point margin in comparison to RoBERTa-10M.”
> - 4. Yes, the difference between Wikipedia and Combined is that Combined contains two wikipedia of the same number of sentences. Wikipedia contains 16M tokens and Combined contains about 31M tokens. We will include this information in the final version of the paper.

---

### Official Review · Reviewer_bCR8 · 2023-08-15

**Soundness:** 3

**Excitement:**

2: Mediocre: This paper makes marginal contributions (vs non-contemporaneous work), so I would rather not see it in the conference.

**Paper Topic And Main Contributions:**

This paper evaluates the capabilities of several variants of BabyBERTa (Huebner et. al., 2021), a smaller RoBERTa architecture trained on a 5M token corpus, on three semantic role labeling/meaning representation datasets, two of which are extractive QA-driven. It analyses the effect of the corpus type, vocabulary size and use of "unmasking" during BabyBERTa pre-training, showing that task performance is generally better when unmasking is not employed and BabyBERTa's smaller vocabulary is used rather than RoBERTa's larger one. The authors also consider the effect of continued pre-training of BabyBERTa, showing that this leads to an improvement in downstream task performance, and the effect of using or not using unmasking during the original and continued pre-training phases.

**Questions For The Authors:**

- (A) What was the motivation for this work, and what is the wider significance of your findings? What insights do they offer into linguistics or efficient NLP systems?
- (B) How did you conduct the significance test mentioned on line 236? Were there multiple training runs for each configuration in Table 5?

**Reasons To Accept:**

- The paper is well written.
- The experimentation seems thorough and mostly supports the authors' claims.
- Replicates the findings of Huebner et. al. on new tasks, increasing confidence in the generality of BabyBERTa's linguistic abilities.

**Reasons To Reject:**

The high-level motivation of this work was not clear to me, and I didn't feel that I learned much from this paper that wasn't already discovered by Huebner et. al. (2021), who introduced BabyBERTa. Some justification is required for the focus on improving BabyBERTa's performance - as I understand it, BabyBERTa was intended primarily as an experiment in child language acquisition research, not the basis for building practical NLP systems. However this is somewhat outside my area of expertise and I might be persuaded by a convincing answer to my question (A).

While Section 3.1 extends Huebner et. al.'s analysis to new tasks and shows that a smaller vocabulary is advantageous in the BabyBERTa training regime, it doesn't offer much in the way of new insight - Huebner et. al. already showed the benefit of unmasking removal. Sections 3.2 & 3.3 attempt to improve BabyBERTa's performance primarily through continued pre-training, but again I don't feel it's very insightful to show that a model trained on 5M tokens performs better when it's trained on 100M or 500M tokens.

There isn't much differentiation in the numbers in Tables 4 & 5 across BabyBERTa settings so I'm not sure we can conclude much about the effect of corpus and masking policy during continued pre-training on downstream performance. The authors claim to have obtained a significant result with a t-test when comparing the performance of BabyBERTa-CHIL on QAMR with and without URP, however they have not explained how they perform this t-test - there is no mention of multiple training runs, so presumably the t-test is over the examples in the test set, so only establishes that one specific training run is better than another, not that one training method is better than another in general. Also, this result seems to be cherry-picked as the authors have not reported significance test results for any of the other configurations and tasks in Table 5. The fact that significance is attained on just one specific configuration and task isn't very meaningful, and I note that on the same task where the authors claim a significant advantage for URP, no-URP works better with the other pre-training corpus.

**Reproducibility:**

4: Could mostly reproduce the results, but there may be some variation because of sample variance or minor variations in their interpretation of the protocol or method.

**Reviewer Confidence:**

3: Pretty sure, but there's a chance I missed something. Although I have a good feel for this area in general, I did not carefully check the paper's details, e.g., the math, experimental design, or novelty.

**Typos Grammar Style And Presentation Improvements:**

There are a few typos (e.g. "pertaining" on lines 200-201). The headings "BabyBERTa" and "RoBERTA" in Table 8 are the wrong way round.

---

> ### Author Rebuttal · Authors · 2023-08-29
>
> Thank you for your valuable comments.
>
> Q(A): BabyBERTa [1] showed an efficient way to pretrain a language model so it can learn grammaticality to the same level as RoBERTa, while using much less training data and fewer parameters. The aim of this paper is to investigate the potential of this efficient language learning - either using BabyBERTa directly or as a starting point - for building NLP systems for downstream tasks, with less data and smaller models. Our work shows that the small language models which gain outstanding grammar capability also achieve promising performance on SRL related tasks after fine-tuning or continually pre-training. We consider this work helps to deepen the understanding of small language models trained on small amounts of data, which has two main implications.
> 1) Both pre-training time and fine-tuning time of BabyBERTa decrease drastically in comparison to RoBERTa-base due to the smaller architecture. Evaluating and understanding the usage and limitations of small language models on downstream tasks can offer valuable insights to guide the development of efficient and environmentally friendly NLP systems in the future.
> 2) Our finding enhances the potential to use transformer-based language models in domains and languages where less data are available.
>
> Q(B): We perform a t-test by comparing the results of two systems. Specifically, we collect the binary results from two different systems on the same test dataset (correct prediction 1, wrong prediction 0) and run a one-tail t-test for it. We did not conduct multiple runs for the configurations in Table 5. However, we report the mean value of three runs of BabyBERTa-CHILDES in the attached table. And we also use the methods recommended in [2] for F1 score evaluation: apply bootstrap to perform the significance test of 3 runs and get the p-value of 0.03 when alpha=0.05. It again shows that the BabyBERTa-CHILDES gains from unmasking removal policy for QAMR. We will add results for multiple runs of all configurations in Table 4&5 in the final version. Our conclusion emphasizes on BabyBERTa benefits from URP on downstream tasks for some starting point datasets such as CHILDES after continual training but not in all configurations.
>
> | | URP\_S | URP\_C |  SRL| QASRL | QAMR |
> |----|----|----|----| ----|-----|
> |BabyBERTa-CHILDES | no |no | 78.04 | 90.48 | 77.60 |
> | | yes |no | 78.08 | 90.43 | 77.88 |
> | | yes |yes | **78.19** | **90.56** | **78.60** |
>
> Presentation Improvements: Thank you for pointing those out and we will fix them in the final version.
>
> [1] Philip A. Huebner, Elior Sulem, Fisher Cynthia, and Dan Roth. 2021. *BabyBERTa: Learning More Grammar With Small-Scale Child-Directed Language.* In Proceedings of the 25th Conference on Computational Natural Language Learning, pages 624–646, Online. Association for Computational Linguistics.
>
> [2] Rotem Dror, Gili Baumer, Segev Shlomov, and Roi Reichart. 2018. *The Hitchhiker’s Guide to Testing Statistical Significance in Natural Language Processing.* In Proceedings of the 56th Annual Meeting of the Association for Computational Linguistics (Volume 1: Long Papers), pages 1383–1392, Melbourne, Australia. Association for Computational Linguistics.

---

### Meta-Review · Area_Chair_UE3M · 2023-09-25

**Recommendation:** 3

**Metareview:**

This paper is focused on evaluating BabyBERTa, a pre-existing language model trained on small-scale child-directed speech, on a series of SRL-based downstream tasks. While the goal of training smaller models on fewer data is potentially useful to the research community, this paper partly suffers from a lack of novelty, as many of its ablations (such as unmasking removal) were already present in Huebner et al.'s original BabyBERTa paper, albeit not evaluated on tasks requiring fine-tuning. Moreover, this work would benefit from running statistical significance tests for all different model configurations to obtain more compelling evidence from the results. Overall, this work is incremental and partly limited in scope and applicability (studying just a single LLM on SRL tasks); on the other hand, it provides some interesting insights into optimal pre-training strategies in low-resource scenarios (such as smaller vocabulary sizes and unmasking removal). Hence, this paper might be considered for Findings.

---

### Decision · Program_Chairs · 2023-10-07

**Decision:**

Accept-Main

**Comment:**

This paper is focused on evaluating BabyBERTa, a pre-existing language model trained on small-scale child-directed speech, on a series of SRL-based downstream tasks. While the goal of training smaller models on fewer data is potentially useful to the research community, this paper partly suffers from a lack of novelty, as many of its ablations (such as unmasking removal) were already present in Huebner et al.'s original BabyBERTa paper, albeit not evaluated on tasks requiring fine-tuning. Moreover, this work would benefit from running statistical significance tests for all different model configurations to obtain more compelling evidence from the results. Overall, this work is incremental and partly limited in scope and applicability (studying just a single LLM on SRL tasks); on the other hand, it provides some interesting insights into optimal pre-training strategies in low-resource scenarios (such as smaller vocabulary sizes and unmasking removal). Hence, this paper might be considered for Findings.